# Vacuum Energy, the Casimir Effect, and Newton's Non-Constant

Benjamin Koch [1,2,3,*], Christian Käding [1], Mario Pitschmann [1] and René I. P. Sedmik [1]

1    Atominstitut, TU Wien, Stadionallee 2, 1020 Vienna, Austria; christian.kaeding@tuwien.ac.at (C.K.);
     mario.pitschmann@tuwien.ac.at (M.P.); rene.sedmik@tuwien.ac.at (R.I.P.S.)
2    Inst. für Theoretische Physik, TU Wien, Wiedner Hauptstrasse 8-10, 1040 Vienna, Austria
3    Instituto de Física, Pontificia Universidad Católica de Chile, Casilla 306, Santiago 782-0436, Chile
*    Correspondence: benjamin.koch@tuwien.ac.at

**Abstract:** The idea of quantum mechanical vacuum energy contributing to the cosmological vacuum energy density is not new. However, despite the persisting cosmological constant problem, few investigations have focused on this subject. We explore the possibility that the quantum vacuum energy density contributes to the (local) gravitational energy density in the framework of a scale-dependent cosmological constant $\Lambda$ and Newton's constant $G$. This hypothesis has several important consequences, ranging from quantum scale-dependence to the hypothetical prospect of novel experimental insight concerning the quantum origin of cosmological energy density.

**Keywords:** quantum vacuum; scale dependence

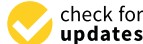



## 1. Introduction

### 1.1. Vacuum Energy and the Cosmological Constant (Problem)

Our modern understanding of physics is based on general relativity (GR) and quantum mechanics, which have evolved into the Standard Model of particle physics (SM) and the cosmological standard model $\Lambda$-CDM. GR is exquisitely tested on various length scales and considered to be the standard theory describing gravity as well as the dynamical evolution of our Universe as a whole.

Einstein amended the theory of GR using a so-called cosmological constant $\Lambda$ to counter the inverse pressure generated by gravity leading to a deflating universe. His aim was to achieve a static universe as a solution of GR, in accordance with common experimental evidence at the time [1]. In 1929, Hubble discovered that the Universe expands [2]—in contrast to the original prediction of GR. Due to this prediction, and lacking further evidence, it was therefore assumed that the Universe was in a decelerating expansion. Einstein rejected $\Lambda$ again, as it was not needed anymore, and also did not yield further insight. Much later, the study of type Ia supernovae led to the surprising discovery that the expansion was actually accelerating [3–5], thereby reviving the need for a positive pressure—and the cosmological constant. Since these first observations, the accelerated expansion has been confirmed with increasing confidence [6–8], which demands for a precisely tuned $\Lambda_0 = 1.1 \times 10^{-52}\,\mathrm{m}^{-2}$ to be included in the field equations of GR. We may interpret $\Lambda_0$ as an energy density contributing via Newton coupling $G_0$ and the speed of light $c$ to the matter part of Einstein's field equations, leading to

$$\rho_{\Lambda_0} = \frac{\Lambda_0 c^4}{8\pi G_0} = 5.35 \times 10^{-10}\,\mathrm{J/m}^3. \tag{1}$$

While GR amended by a cosmological constant allows one to describe the accelerated cosmic expansion, the interpretation of this constant in terms of vacuum energy [9] leads to a severe fine-tuning problem [10,11]. This is based on the fact that zero-point energies of all quantum fields, i.e., those of the SM as well as possibly other yet unknown ones, provide additional contributions to Einstein's original (bare) cosmological constant. Furthermore,

the Higgs potential leads to an effective contribution during its phase transition related to electroweak symmetry breaking [12]. While the measured value of $\Lambda_0$, given above, necessarily contains all of these contributions, the zero-point energies of each particle's quantum field provide infinite contributions to $\Lambda_0$, which are ordinarily subtracted via operator normal ordering. However, this subtraction would render $\Lambda_0 = 0$. If, instead, these contributions are kept and rendered finite by introducing a momentum cutoff $\kappa_0$ at the electroweak scale $m_Z$, one obtains values that are some 55 orders of magnitude above $\Lambda_0$ [11]. For a cutoff chosen at the Planck scale, the mismatch would even be 120 orders of magnitude [10]. This problem can be expressed as a dimensionless ratio of energy densities. The energy density corresponding to a momentum cutoff would be $\rho_Q \sim c\kappa_0^4/\hbar^3$, which can be set in relation to the observed energy density (1)

$$\Upsilon_0 \equiv \frac{\rho_{\Lambda_0}}{\rho_{Q,0}(\kappa)} = \frac{\Lambda_0 c^3 \hbar^3}{8\pi G_0 \kappa_0^4} = \begin{cases} 10^{-121} & \text{for } \kappa_0 = c\sqrt{\frac{c\hbar}{G_0}} \\ 10^{-55} & \text{for } \kappa_0 = cm_Z. \end{cases} \tag{2}$$

Thus, according to our current understanding, the value of $\Upsilon_0$ should be of order one (or exactly zero). Instead, the values in Equation (2) are far too small. Such 'miraculous' reductions by many orders of magnitude leading to a small but non-zero value of $\Lambda_0$ after the summation of all individual contributions constitute the cosmological constant problem (CCP) [10,11]. Similar as for the strong $CP$ problem of quantum chromodynamics, the CCP is a severe fine-tuning problem, demanding explanation. While some argue using the anthropic principle that $\Lambda_0$ could be a constant of nature determined coincidentally by the fact that we exist to ask for its origin [13,14], this point of view may appear rather metaphysical and unsatisfactory. A physical explanation, however, seems to be beyond the present SM and GR framework, which creates a strong case for our current understanding of the cosmic evolution and the quantum vacuum being incomplete.

Consequently, this problem gave motivation for research on possible alternative descriptions of quantum gravity (QG). The list of approaches and techniques is long, ranging from standard quantum field theory formulations, such as the functional renormalization group approach [15–18], to Planck scale fluctuations [19,20], holographic interpretations [21,22], and many more.

In any case, new insight (particularly from experiments) concerning the quantum origin of $\Lambda_0$, Equation (1), is urgently needed. Such insight, however, is not limited to arise from astronomical measurements but can also be gained in the laboratory with carefully chosen observables. This paper establishes such a possible link between laboratory experiments and quantum effects of gravity.

### 1.2. Vacuum Energy in the Casimir Effect

Another physical context, where (quantum) fluctuations play a pivotal role, is the Casimir effect. Derived in 1948 by Hendrik Casimir [23] and quantitatively confirmed[1] nearly half a century later [27], this effect remains the only known experimental manifestation of the quantum vacuum causing a detectable interaction between macroscopic bodies. The Casimir effect is caused by spatial boundary conditions that limit the spectrum of quantum fluctuations. This effect can be calculated for all sorts of fields and interactions, but we will focus on electromagnetic fluctuations, since these are most directly accessible for experimental and phenomenological studies [28]. In fact, the electromagnetic interaction is the only one for which Dirichlet-like boundary conditions can technically be realized. In the volume enclosed between the boundaries, the mode density is changed[2], resulting in a force between the boundaries. The difference between the energy of a free vacuum and the restricted case is called the Casimir energy $E_C$. For the ideal case of two infinitely extended, perfectly conducting parallel plates at distance $a$, one obtains for the energy per unit area $A$ [23]

$$\frac{E_{C,\text{id}}(a)}{A} = -\frac{\hbar c \pi^2}{720 a^3}, \tag{3}$$

where $E_C = \int \mathrm{d}V \rho_C(\mathbf{r}) = \int \mathrm{d}V \langle 0|T_{00}|0\rangle_{\mathrm{ren}}$ is formally obtained from the renormalized vacuum expectation value of the time–time component of (electromagnetic) energy momentum tensor $T_{\mu\nu}$, integrated over the volume $V$ between the boundaries. Again, renormalization is performed via subtraction of the energy densities $\rho_C$ for the boundaries in the actual configuration, and at infinite separation [26]. $E_C$ can be computed straightforwardly for plane and curved geometries, taking into account spectral dielectric properties, roughness, temperature, and other experimental details using semi-classical Lifshitz theory [35] or a range of more modern methods (review: [26,36,37]). Real materials show dispersion and their response to electromagnetic fields typically falls off $\propto \omega_p/\omega^{-2}$ for high frequencies $\omega$ and the plasma frequency $\omega_p$, eventually leading to transparency in the far UV. Since the Casimir interaction depends on a spectrum of modes, we can also associate an effective penetration depth $\delta_p \sim c/\omega_p$ with a material, which approximately quantifies the reach of the Casimir energy into the material. For metals, $\lambda_p$ is of the order of $100\,\mathrm{nm}$. However, while the computation of the Casimir energy is well understood, a local measure for this energy in the form of energy density $\rho_C$ poses a problem [38,39]. The idealistically assumed step in the dielectric functions between the gap and the bounding surface leads to divergences $\propto z^{-4}$, depending on the distance $z$ from the (Dirichlet) boundary [40,41]. One possible solution is the introduction of a soft [40,42,43] or movable [44–47] wall that smooths out the discontinuity in the dielectrc properties and thereby eliminates divergences [48]. However, lacking a detailed first-principles approach for the material at frequencies $> c/\ell$ with $\ell$ being the typical length scale of interatomic spacings, quantitative predictions will strongly depend on the assumptions taken. The same is true for the region inside the walls, for which no analytic form has been found and only numerical methods on the basis of ad hoc assumptions for a smooth potential lead to finite values for $\rho_C$ [49]. Independent of these details, however, $\rho_C$ can be expected to be large near the boundaries, for which it could potentially contribute to gravitational interactions [50].

Up to now, experiments have only tested configurations in which the Casimir interaction is several orders of magnitude stronger than the gravitational interaction. The upcoming CANNEX setup [51], however, will for the first time demonstrate a regime in which the Casimir and gravitational interactions are similar in strength. This opens up the exciting possibility of testing gravity in physical conditions with modified vacuum energy density.

### 1.3. Hypotheses

In the previous subsections, we discussed the cosmological vacuum energy density $\rho_{\Lambda_0}$ in Equation (1) and the quantum-laboratory energy density $\rho_C$. Despite their suggestively analogous names, it is not at all clear whether these two quantities are related at all (for a review, see [12,52]). The seriousness of our lack of theoretical understanding of this relation is exemplified by the CCP [c.f. Equation (2)].

In this paper, we show how one can experimentally test the hypothesis that both energy densities are related. The resulting modified cosmological energy density $\rho_\Lambda$ will then be a function of the original cosmological energy density $\rho_{\Lambda_0}$ and the quantum contribution $\rho_C$. In general, we have no clear indication of what this function could look like. However, given that measurable variations of the Hubble constant [53] that could be interpreted (in many ways) as a variability of $\Lambda$ are at the 10% level, it appears reasonable to assume a small linear change in $\rho_\Lambda$ by $\rho_C$,

$$
\begin{aligned}
\rho_\Lambda &= \rho_\Lambda(\rho_0, \rho_C) \\
&\approx \rho_{\Lambda_0} - \alpha \rho_C.
\end{aligned}
\tag{4}
$$

Here, $\alpha$ is the free parameter of this hypothesis. The sign of this parameter is not determined, and thus the sign in Equation (4) is pure convention. Since the Casimir energy density arises from electromagnetic vacuum fluctuations, which in turn can be expected to contribute fully to the total energy density, even extreme values, such $|\alpha| = 1$, are possible. With

this reasoning, a cosmological model [54–56] was built and measurable consequences of a common cutoff have been discussed [57]. On the other hand, it has been shown that $\rho_C$ actually arises from loop diagrams with external legs as opposed to 'true' vacuum bubbles without legs [58]. From this perspective, it can be conjectured that maybe $\rho_\Lambda$ arises exclusively from a topologically different type of diagram, which would imply $\alpha = 0$. Thus, a laboratory test of hypothesis (4) would give valuable information with this respect.

Any value $\alpha \neq 0$ in relation (4) has notable consequences. Since the Casimir energy density is a function of external dimensionful quantities and local coordintes $\vec{x}$, the same must be true for the resulting $\rho_\Lambda$. Due to this dependence on local and global scales $k \equiv k(\vec{x})$, we call it a scale-dependent (SD) quantity, $\rho_\Lambda = \rho_\Lambda(k)$. Consequently, the definition in terms of the gravitational couplings (1) has to be generalized to allow for scale-dependence

$$\rho_\Lambda(k) := \frac{c^4 \Lambda(k)}{8\pi G(k)} \ .$$ (5)

Thus, the gravitational couplings are also generalized to SD quantities $G = G(k)$ and $\Lambda = \Lambda(k)$ [59–62]. Such couplings have to be dealt with in a theoretical framework, known as SD gravity. It will be introduced in the next subsection.

Note that in the recent literature there have been numerous discussions on the interplay between the Casimir force and (quantum) effects of gravitational interactions [63–67]. These are not related to the scale-dependent scenario discussed in our paper since they are either limited to a particular proposal of enhanced interactions between quantum fluids and gravity or are purely idealized formal discussion.

*1.4. Scale Dependence, Quantum Fluctuations, and Scale Setting*

The equations of motion in classical physics are only an approximation of the underlying laws of quantum field theory. In quantum field theory, this is reflected by the fact that all couplings $\alpha$ are, in fact, subject to a renormalization scale $k$: $\alpha \to \alpha(k)$. The functional form of these couplings $\alpha(k)$ is determined by the renormalization group equations [68,69]. The 'classical' value of these couplings $\alpha_0$ typically corresponds to some infrared limit of these couplings, e.g., $\alpha_0 = \lim_{k \to 0} \alpha(k)$.

In the effective field theory approach to quantum gravity [15–18,70,71], one works with the effective action of quantum gravity coupled to matter, which in the local low curvature expansion can be written as

$$\Gamma_k = \int d^4 x \sqrt{g} \left( \frac{R - 2\Lambda(k)}{G(k)} + \mathcal{L}_m(\phi, k) \right),$$ (6)

where the scale-dependent effective Lagrangian of matter fields $\phi$ is denoted by $\mathcal{L}_m(\phi, k)$. Note again that in Equation (6), both gravitational couplings, the cosmological and the Newtonian ones, are assumed to be scale-dependent.

$$\Lambda \to \Lambda(k), \quad \text{and } G \to G(k).$$ (7)

The commonly used measured values of these couplings are then associated with the asymptotic (classical) values in the far infrared,

$$\Lambda_0 = \lim_{k \to 0} \Lambda(k), \quad \text{and } G_0 = \lim_{k \to 0} G(k).$$ (8)

Even though a uniquely accepted running of gravitational couplings is not available, we can extract useful information from the SD of gravitational couplings in the close vicinity of a given IR scale $k_0 \ll M_P$. For this purpose, we can expand the SD gravitational couplings around this scale [72,73]

$$G(k) = G_0(1 + g(k)) = G_0\left(1 + C_1 G_0 k^2 + C_2 G_0^2 k^4\right) + \mathcal{O}(k^4), \tag{9}$$

$$\Lambda(k) = \Lambda_0(1 + \lambda(k)) = \Lambda_0\left(1 + C_3 G_0 k^2 + C_4 G_0^2 k^4\right) + \mathcal{O}(k^4). \tag{10}$$

Here, the $C_i$ parametrizes the first effects of running couplings, when we depart from the classical limit. Note that Equation (6) is valid only for a theoretical effective field description, which is the result of a regularization and renormalization procedure. Therefore, all appearing coupling parameters $G_0$, $\Lambda_0$, $C_i$, ... are also effective parameters and not bare ones, and all quantities appearing in Equations (4) and (5) are renormalized ones. In this sense, the normalization condition in Equations (9) and (10) is such that $G_0$ and $\Lambda_0$ are the "classical" couplings which are typically measured in the absence of quantum effects at astrophysical or cosmological scales. Even quite different approaches to quantum gravity (e.g., [74]) can, in principle, be mapped to Equations (9) and (10), but they would predict different values for the coupling parameters $C_i$. In the appendix, we derive values for $C_1$ and $C_3$, as predicted by asymptotic safety coupled to matter. These benchmark values, given in Table A1, are of order one. Subsequently, however, we will treat $C_i$ as free parameters and explore whether and how they could be tested experimentally. From a pure power counting argument, we would expect these parameters to be too small to be experimentally observable in the near future. Though, due to infrared instabilities in the renormalization group flow, these parameters could become largely enhanced [75]. Anyway, to our knowledge there existed no experimental constraints on the $C_i$ until now.

To extract quantitative predictions from SD couplings like Equations (9) and (10), we need to set the RG scale $k$ in terms of physical parameters '$(t, x, a, \dots)$' of a given observation (or experiment) that can be measured, i.e., $k \to k(t, \vec{x}, a, \dots)$. This procedure is known as scale-setting, and is crucial for obtaining measurable predictions. There exists a large variety of scale-setting methods in conventional quantum field theory [76–79] and in quantum gravity [80–89]. For applications concerning the scaling behaviour of the cosmological constant, see Refs. [59–62]. The applicability of these methods strongly depends on the physical and experimental setting. In the context of the aforementioned fine-tuning problem, they play an important role in attempts to solve the cosmological constant problem [57] (see also references [88,90–92]).

Regardless of which quantum gravity model Equations (9) and (10) are obtained from or which scale setting method is used, after the scale setting, the SD couplings can be written as local quantities. Thus, the Einstein field equations only remain consistent if they are generalized for SD couplings [80]. Varying the action (6) with respect to the metric field, the field equations read

$$G_{\mu\nu} = 8\pi \frac{G(k)}{c^4} T_{\mu\nu} - \Lambda(k) g_{\mu\nu} - \Delta t_{\mu\nu}(k), \tag{11}$$

where now

$$\Delta t_{\mu\nu}(k) = G(k)\left(g_{\mu\nu}\nabla^2 - \nabla_\mu\nabla_\nu\right)\frac{1}{G(k)}. \tag{12}$$

If we leave the quantum gravity motivation of the scale-dependent couplings aside, Equation (11) with its covariant form can be interpreted as a particular (minimal) case of the broader class of scalar–tensor theories of gravity [93,94]. Thus, even though the details have to be checked case by case, our findings concerning the interplay between Casimir energy and gravitational couplings may also result from scalar–tensor extensions of general relativity. Another type of theory, in which the field equations take a form similar to Equations (11) and (12), is $f(R)$ gravity [95], where $R$ is the Ricci scalar. However, such a correspondence can only be established if the scale-setting imposes $k^2 \sim R$, which is a common choice for problems concerned with the early Universe (see Section 2.2). We recently used Equation (11) to study black holes, the early Universe, and the phenomenology of the late Universe [73,96–101]. Note that in the literature there is an ongoing discussion of whether gravitating quantum vacuum (Casimir) energy is compatible with energy mo-

mentum conservation and shift symmetry [50,102–104]. For this reason, we comment here shortly on how these two aspects are addressed in the approach of Equation (11):

- In this approach, the principle of equivalence as discussed in the literature is replaced by general covariance of the Einstein tensor $\nabla^\mu G_{\mu\nu} = 0$. This implies that the covariant derivative of the entire right-hand side of Equation (11) vanishes. Thus, a conservation of each individual term, such as $\nabla^\mu T_{\mu\nu} = 0$, is not strictly required anymore.
- If the Casimir energy density $T_C^{\mu\nu}$ contributes to the matter energy density $T^{\mu\nu} = T_M^{\mu\nu} + T_C^{\mu\nu}$, then the corresponding Lagrangian must have a shift symmetry $\mathcal{L} = \mathcal{L}_M + \mathcal{L}_C$. Interestingly, such a shift symmetry can be used as an additional condition, which allows one to directly solve Equation (11). The condition arising from this symmetry applied to Equation (12) is the so-called null energy condition. We introduced and investigated this condition in [87,88,96,99].

In the present article, we neither use the shift symmetry nor the explicit covariance of $T^{\mu\nu}$ for which our findings stand independent of the discussion in the literature [50,102–104].

## 2. Results

We now present a discussion of the implications of Equation (11) and a scale-setting for a Casimir experiment.

### 2.1. Gravitational Field Equations with Weak Curvature and Weak SD

In this subsection, we will derive and explore the Weak Gravitational curvature and Weak SD (WG-WSD) limit of Equation (11). For this purpose, we take the trace of the equations to isolate the Ricci curvature tensor on the left-hand side

$$R^\mu_\nu = 8\pi \frac{G(k)}{c^4}\left(T^\mu_\nu - \frac{1}{2}\delta^\mu_\nu T\right) + \Lambda(k)g^\mu_\nu + G(k)\left(\frac{1}{2}\delta^\mu_\nu \nabla^2 + \nabla^\mu\nabla_\nu\right)\frac{1}{G(k)}. \tag{13}$$

The WG-WSD limit is then obtained via an expansion in formally small corrections to the flat Minkowski solution. Having in mind Casimir experiments with largely non-relativistic dynamics, the line element is assumed to be approximately static ($t$-independent). This line element is expanded with the parameter $\epsilon_\Phi$ and the SD gravitational coupling is expanded with the parameter $\epsilon_G$. For both bookkeeping parameters ($\epsilon_\Phi, \epsilon_G$), we consider

$$
\begin{aligned}
ds^2 &= -(1 + 2\epsilon_\Phi\Phi(r,\theta,\phi))c^2dt^2 + (1 - 2\epsilon_\Phi\Psi(r,\theta,\phi))dr^2 \\
&\quad + (1 + 2\epsilon_\Phi\Xi(r,\theta,\phi))r^2d\Omega^2 + \mathcal{O}(\epsilon_\Phi^2), \\
G(k) &= \epsilon_\Phi\left(G_0 + \epsilon_G\Delta G(k) + \mathcal{O}(\epsilon_\Phi^2)\right), \\
\Lambda(k) &= \epsilon_\Phi\left(\Lambda_0 + \epsilon_G\Delta\Lambda(k) + \mathcal{O}(\epsilon_\Phi^2)\right),
\end{aligned}
\tag{14}
$$

where $(\Phi, \Psi, \Xi) \ll 1$ are the gravitational potentials and $\Delta G \ll G_0$ is the SD correction to the gravitational coupling $G_0$. Further, for non-relativistic matter at rest with energy density $\rho_M = \rho_M(r,\theta,\phi)$, the stress energy tensor (e.g., in spherical coordinates) reads simply

$$\left(T^\mu_\nu - \frac{1}{2}\delta^\mu_\nu T\right) = \frac{\rho_M}{2}\mathrm{diag}(-1,1,1,1). \tag{15}$$

With this matter contribution, and the expansion (14), the time–time component of Equation (13) reads

$$\vec{\nabla}^2\Phi(r,\theta,\phi) = \frac{4\pi}{c^4}G_0\rho_M(r,\theta,\phi) + \frac{\epsilon_G}{\epsilon_\Phi}\frac{\vec{\nabla}^2\Delta G(k)}{2G_0} - \Lambda(k) + \mathcal{O}(\epsilon_\Phi, \epsilon_G), \tag{16}$$

where a global factor of $-\epsilon_\Phi$ has been canceled out. This expansion is valid and physically reasonable only in a regime where $\epsilon_\Phi > \epsilon_G > \epsilon_\Phi^2 > 0$. Note that we also expect

contributions $\propto \Delta G(k) \rho_M$ to Equation (16). Such contributions do exist, but they are much smaller than the leading contributions shown in Equation (16). Now, we drop the formal expansion parameters, keeping in mind the smallness of the $\Delta G$ contribution. For all practical purposes in the context of Casimir experiments, the cosmological term can be numerically neglected with respect to the other terms. Similarly, the energy density that the Casimir effect contributes to $\rho_M$ can be neglected for most experimental purposes. However, there is a specially designed experiment [105–107] which will provide insight on the question of whether "gravity ignores vacuum energy" [104]. For this particular experiment $\rho_M$ will include $\rho_C$ as a small additive factor. Thus, the correction to the gravitational coupling is known and given in terms of the local Casimir energy density $\Delta G(k) = \Delta G(\rho_C(\vec{x})) \equiv \Delta G(\vec{x})$ (as it will be shown in the next sections). Thus, the Poisson equation for a general gravitational potential in the WG-WSD limit (16) can be solved with the usual Green's function method

$$
\begin{aligned}
\Phi(\vec{x}) &= \frac{G_0}{c^2} \int_{V_1} d^3 x' \frac{\rho_M(\vec{x}') + c^2 \frac{\vec{\nabla}^2 \Delta G(\vec{x}')}{8\pi G_0^2}}{|\vec{x} - \vec{x}'|} + \mathcal{O}(\epsilon) \\
&= \frac{G_0}{c^2} \int_{V_1} d^3 x' \frac{\tilde{\rho}_M(\vec{x}')}{|\vec{x} - \vec{x}'|} + \mathcal{O}(\epsilon),
\end{aligned}
\tag{17}
$$

where $V_1$ is the region of the gravitational source and we defined the apparent gravitational energy density

$$
\tilde{\rho}_M = \rho_M + c^4 \frac{\vec{\nabla}^2 \Delta G}{8\pi G_0^2}.
\tag{18}
$$

Now, the result (17) can be inserted into the geodesic equation for a test particle with position $x^\mu$

$$
\frac{d^2 x^\mu}{ds^2} + \Gamma^\mu_{\alpha\beta} \frac{dx^\alpha}{ds} \frac{dx^\beta}{ds} = 0.
\tag{19}
$$

We find, for the spatial components, in the non-relativistic limit

$$
\frac{d^2 \vec{x}}{dt^2} = -c^2 \vec{\nabla} \Phi + \mathcal{O}(\epsilon_\Phi).
\tag{20}
$$

To relate this acceleration to a force in Newton's second law, we require a mass. This mass should be given by a volume integral over a source mass density. For the latter, we have two options at our disposal:

- The 'original' mass density $\rho_M/c^2$, which is typically determined with electromagnetic forces, gauged in the absence of the Casimir effect. The integral over this density is the mass we find in the absence of $\Delta G$: $M_2 = \int_{V_2} d^3 x \, \rho_M(x)/c^2$. However, this is not the density that allows one to define a force for the gravitational acceleration from Equation (20) since it does not fulfil Newton's third law.
- The apparent gravitational mass density (18), which appears in the gravitational potential (17). The gravitational force caused by one object on another object $\vec{F}_{12}$ must obey Newton's third law $\vec{F}_{12} = -\vec{F}_{21}$. In the WG-WSD limit, this is only guaranteed for the apparent gravitational mass density $\tilde{\rho}_M/c^2$, for which this quantity must appear in the definition of gravitational force.

We thus find that the force sensed by an extended body with volume $V_2$ is given to the leading order in $(\epsilon_G, \epsilon_\Phi)$ by

$$
\begin{aligned}
\vec{F}_{G,12} = -\vec{F}_{G,21} &= -\int_{V_2} d^3 x_2 \, \tilde{\rho}_M(\vec{x}_2) \vec{\nabla} \Phi(\vec{x}_2) + \mathcal{O}(\epsilon_\Phi) \\
&= \frac{G_0}{c^4} \int_{V_2} d^3 x_2 \int_{V_1} d^3 x_1 \frac{\tilde{\rho}_M(\vec{x}_1) \tilde{\rho}_M(\vec{x}_2)(\vec{x}_2 - \vec{x}_1)}{|\vec{x}_2 - \vec{x}_1|^3} + \mathcal{O}(\epsilon_\Phi).
\end{aligned}
\tag{21}
$$

The crucial difference between Equation (21) and the usual expression for the gravitational force between extended bodies is the distinction between $\tilde{\rho}_M$ and $\rho_M$, which only appears for $\vec{\nabla}^2 \Delta G \neq 0$. In the following section, we will derive the functional form of $\Delta G$ for Casimir experiments.

### 2.2. Scale Setting

To relate an effective action to real observables, one has to choose the RG scale in terms of variables that describe the system under consideration. Since these variables are in many cases local, the scale-setting can imply the breaking of local symmetries, unless one takes particular care to maintain symmetry throughout the scale-setting procedure.

This necessary procedure of scale-setting is an open problem in theories with quantum effects and gravity. Naturally, it is also a source of large theoretical uncertainties because there is a large number of possibilities of how it is to be implemented in practice [59–62,82,108–117]. In principle, different scale-settings give different predictions. Luckily, the degree of uncertainty reduces drastically if one is only concerned with weak scale-dependence, as in this study. In this case, different scale-settings lead to the same prediction.

Below, we show how a scale-setting based on the definition of density and a manifestly covariant scale setting both lead to the same type of expression.

#### 2.2.1. Density-Induced Scale-Setting

Since we are interested in the leading corrections to the asymptotic limit (8), we insert Equations (9) and (10) into the definition (5) and expand to the first order in $k^2$

$$\rho_\Lambda(k) = \rho_{\Lambda_0} - k^2 c^4 \frac{(C_1 - C_3)}{8\pi} \Lambda_0 + \mathcal{O}(k^4). \tag{22}$$

Here, the second term is the aforementioned correction to the asymptotic definition (1). By construction, Equation (22) is equal to $\rho_\Lambda$, defined in Equation (4). Thus, by subtracting Equations (4) and (22), we obtain the unique scale setting that is consistent with the working hypothesis in Equation (4),

$$k^2 = \alpha \frac{8\pi \rho_C}{c^4(C_1 - C_3)\Lambda_0}. \tag{23}$$

Reinserting this into the IR expansion (9), we find that the gravitational coupling inherits a weak dependence on the electromagnetic Casimir energy density

$$G(k) \approx G_0 \left( 1 + C_1 G_0 \alpha \frac{8\pi \rho_C}{c^4(C_1 - C_3)\Lambda_0} \right). \tag{24}$$

Comparing Equation (24) with the weak SD expansion in Equation (14), we can identify

$$\Delta G \approx \alpha G_0^2 \frac{8\pi C_1 \rho_C}{c^4(C_1 - C_3)\Lambda_0}. \tag{25}$$

This correction has to be inserted into Equation (17) when calculating the modified gravitational potential or the induced force between two objects according to Equation (21). Consequently, the three phenomenological parameters in the following discussion will be $(\alpha, C_1, C_3)$.

#### 2.2.2. Explicitly Covariant Scale-Setting

In terms of the action (6), the cosmological constant problem becomes the question "how and to which extend do quantum modes of $\mathcal{L}_m(\phi, k)$ contribute to $\Lambda(k)$". Varying Equation (6) with respect to the metric field $g_{\mu\nu}$ gives rise to field Equation (11). The general covariance of the system, even after the scale setting, can be assured by the variational scale-setting prescription [83], which complements (11) with the condition

$$\frac{\partial}{\partial k^2}\left(\frac{R-2\Lambda(k)}{G(k)}+\mathcal{L}_m(\phi,k)\right)=0. \tag{26}$$

This condition is typically hard to solve, but since we are interested in IR effects we can expand the matter Lagrangian in analogy to Equations (9) and (10)

$$\mathcal{L}_m(\phi,k)=\mathcal{L}_{m,0}(\phi)+k^2\mathcal{L}_{m,1}(\phi)+\dots \tag{27}$$

Note that for a given quantum field theory the corrections $\mathcal{L}_{m,1}$ to the classical Lagrangian couplings $\mathcal{L}_{m,0}(\phi)$ can be calculated explicitly. For our purposes, however, it is sufficient to know that they exist. Inserting the IR expansions (9), (10), and (27) into the covariant scale-setting condition (26), we can solve the resulting relation for $k^2$ and call this solution the "optimum" scale

$$k_{opt}^2=\frac{\mathcal{L}_{m,1}(\phi)+2(C_1-C_3)\Lambda_0}{4G_0\Lambda_0(C_1^2-C_2-C_1C_3+C_4)}\approx\frac{\mathcal{L}_{m,1}(\phi)+2(C_1-C_3)\Lambda_0}{4G_0\Lambda_0(C_1^2-C_1C_3)}, \tag{28}$$

where we neglected contributions of order $R$ and higher orders of $(C_2, C_4)$. The leading local contribution to this optimum scale comes from the matter and field Lagrangian. For the case of the electromagnetic field, the leading Lorentz invariant terms of this effective Lagrangian are [118]

$$\mathcal{L}_{m,0}=\frac{1}{2}\left(\vec{E}^2+\vec{B}^2\right)+a\left(\vec{E}^2+\vec{B}^2\right)^2+a^*(\vec{E}\cdot\vec{B})^2, \tag{29}$$

$$\mathcal{L}_{m,1}=a_1\left(\vec{E}^2+\vec{B}^2\right)+a_2\left(\vec{E}^2+\vec{B}^2\right)^2+a_1^*(\vec{E}\cdot\vec{B})^2. \tag{30}$$

The coupling coefficients $(a, a^*, a_1, a_2, a_1^*)$ are the calculable coefficients of the underlying quantum field theory (dominated by quantum electrodynamics). All $a_i$, $a_{i*}$ vanish in the classical limit. The fields $\vec{E}$ and $\vec{B}$ are the local average of the quantum electric and magnetic field. We could insert these expansions into the optimum scale (28) in this invariant form, but the expression simplifies when we restrict the following discussion to the case of the electromagnetic modes of the Casimir effect, where we have

$$\left(\vec{E}^2+\vec{B}^2\right)=2\rho_C(x). \tag{31}$$

Thus, for this experimental configuration, we can write

$$\mathcal{L}_{m,1}(\phi)=\mathcal{L}_{m,1}|_{const.}+\tilde{\alpha}\cdot\rho_C(x)+\mathcal{O}\left(\rho_C^2\right), \tag{32}$$

where $\tilde{\alpha}$ is the corresponding proportionality factor. Inserting this into Equation (28), we find that the optimum scale is again a global constant ($\tilde{k}_0^2$) plus a linear contribution from the Casimir vacuum energy density

$$k_{opt}^2=\tilde{k}_0^2+\tilde{\alpha}\frac{\rho_C}{4C_1(C_1-C_3)G_0\Lambda_0}. \tag{33}$$

If we now relabel the constant parameters of our hypothesis as

$$\tilde{\alpha}=32\frac{C_1G_0\pi}{c^4}\alpha, \tag{34}$$

the local part of the covariant scale-setting corresponds exactly to the previously implied density scale-setting in Equation (23), and the local change in the gravitational coupling is equal to (25).

Note that even though the final result (25) does not look explicitly covariant, there is no reason to worry. We must remember that any local source breaks covariance in GR.

For example, even though the metric solution of the interior of a star does not have all the symmetries of GR in free space, it is still the result of covariant field equations. Analogously, the result (25) is the product of a covariant scale-setting procedure (26) combined with covariant field Equation (11).

### 2.3. Perspectives for Experimental Tests

We now attempt to find ways to experimentally test our hypothesis. From the results in Equations (21) and (25), it is clear that any Casimir experiment which is sensitive enough to also measure the gravitational attraction between two objects, such as [51], will be suited to resolve, to some extent, the difference between $\rho_M$ and $\tilde{\rho}_M$. Respective results would allow us to determine or constrain the parameters $\alpha, C_1, C_3$ of our hypothesis. To obtain a glimpse of the possible experimental importance of this effect, it is necessary to make further assumptions about experimental details. For this purpose, we have to integrate Equation (21) over the volume of the two interacting plates of the experiment. Let us consider the situation depicted schematically in Figure 1.

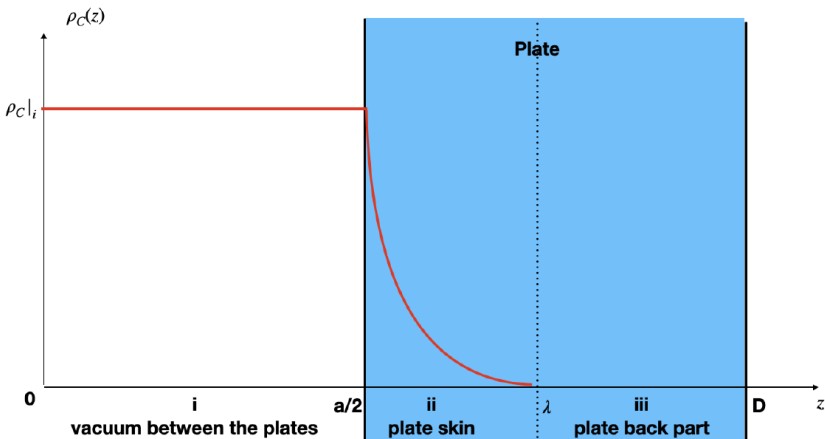

**Figure 1.** Schematic sketch of an assumed exponential attenuation of $\rho_C(z)$ over the skin depth inside of a material, plotted from the middle between two plates at $z = 0$ towards the plate in the region with $z > 0$; the region between the plates is labeled (i), the skin of the plate is labeled (ii), and the region with effectively vanishing $\rho_C$ inside the plate is labeled (iii). The configuration is symmetric under $z \leftrightarrow -z$.

In this configuration, the Casimir vacuum energy density is idealized and assumed to be constant between the plates $\rho_C = \rho_C|_{i)}$ and which is defined by the Casimir free energy $\mathcal{F}_C$ obtained numerically from the Lifshitz (or a more modern) theory under consideration of the experimental parameters [51,119]. Led by physical intuition, we further assume $\rho_{C_{ii)}}$ to drop exponentially to zero when entering the plates

$$\rho_C(z) = \theta(a/2 - |z|) \cdot \rho_C|_{i)} + \theta(|z| - a/2)\rho_C|_{i)} \cdot \exp\left(-\left(|z| - a/2\right)/\delta_p\right). \tag{35}$$

The scale-setting condition (23) remains valid for this scenario, which implies that $G(k)$ is constant wherever $\rho_C$ is constant. Thus, $\Delta t_{\mu\nu}$ vanishes in regions (i) and (iii) in Figure 1. Since, further, the numerical value of $\Lambda(k)$ is negligibly small in all regions, this means that in the regions outside the plate, the usual vacuum Einstein equations apply, and thus Equation (21) can be used with $\vec{\nabla}^2 \Delta G = 0$ there.

Since the above Equation (35) is a bold simplification of a still unknown, but likely more complicated, functional form of $\rho_C(z)$ [41–43,48,49], we defer a detailed numerical integration of Equation (21) and leave this topic for future work. Instead, we revisit the WG-WSD assumption of Equation (14) to obtain a rough estimate of what we could expect to find for the phenomenological parameters. By inspecting the result (21), it is reasonable to assume that the integrated energy density of the matter material $\rho_M$ is bigger than the

correction due to the integral over SD of the gravitational coupling. Thus (to the leading order in $\alpha$),

$$1 \ll (8\pi G_0^2) \frac{\int_{a/2}^{D} \mathrm{d}z\, \rho_M(z)}{\int_{a/2}^{D} \mathrm{d}z\, c^2 |\vec{\nabla}^2 \Delta G|} \approx \frac{\alpha C_1}{C_1 - C_3} \frac{\rho_{C,i}}{c^2 \delta_c \Lambda_0 \rho_M D}. \tag{36}$$

Using Equations (3) and (25) and realistic exemplary values of ($z = a/2$, $a = 10^{-5}$ m, $\rho_M/c^2 = \rho_{\mathrm{gold}}/c^2 = 19.3$ g/cm$^3$, $\delta_c = 10^{-7}$ m, $D = 0.01$ m) in the above expression, we find that

$$\left| \frac{\alpha}{\tilde{\lambda}} \frac{C_1}{C_1 - C_3} \right| \ll 10^{-30}. \tag{37}$$

Note that $\delta_c$ and the boundary value $\rho_{C,i}$ refer to the assumed exponential attenuation model in Equation (35), which is subject to significant theoretical and experimental uncertainty. Nevertheless, as the skin depth was investigated quantitatively in experiments [120], we think that the error in this approach with respect to reality will amount to a mere few orders of magnitude, for which the bound (37) will remain approximately valid even in the worst case. This clearly implies a very strong impact on the parameter space $(\alpha, C_1, C_3)$.

A more realistic modeling of the experimental situation in this and other configurations, such as the measurement of the potential $\Phi(z)$ [c.f. Equation (17)] with test particles, will be part of our future projects.

## 3. Discussion and Conclusions

In the foregoing sections we attempted to derive how novel experimental insight into the possible connection between quantum vacuum energy and energy density corresponding to the cosmological coupling $\Lambda$ could be obtained. Now, we discuss the result of these considerations, Equation (37).

### 3.1. Why So Strong?

The sensitivity in Equation (37) is, despite its large uncertainty due to the simple assumed model for attenuation inside the material, overwhelmingly strong when we compare it with the standard quantum gravity corrections to the Newtonian potential [121]. These leading corrections are typically suppressed as $r_S/a$ or $\lambda_p^2/a^2$, where $r_S$ and $\lambda_p$ are the Schwarzschild radius and the Planck length, respectively, and $a$ is a typical length scale. The Planck length scale is extremely small, such that one would have

$$\frac{\sqrt{\hbar G_0/c^2}}{a} \approx 10^{-30}. \tag{38}$$

Let us say an observable with such a correction is testable to order one, i.e.,

$$\tilde{\alpha} \frac{\sqrt{\hbar G_0/c^2}}{a} \lesssim 1. \tag{39}$$

The phenomenological pre-factor $\tilde{\alpha}$ of such a test could then only be very weakly constrained

$$\tilde{\alpha} \lesssim 10^{+30}. \tag{40}$$

The reason for the discrepancy between the usual expectation, given by Equation (40), and the strength of our result in Equation (37) is fourfold:

(i) Comparable order of $\rho_{\Lambda_0}$ and $\rho_C$.

At experimental scales of $a \approx 5 \cdot 10^{-5}$ m, the cosmological energy density given by Equation (1) and the Casimir energy density are of the same order of magnitude. This means that relative corrections in $\rho_\Lambda$ are not necessarily small, unless $|\alpha| \ll 1$. This is usually not the case in estimates like Equation (40).

(ii) Link $\rho_\Lambda \leftrightarrow \Lambda(k) \leftrightarrow G(k)$.

In our discussion, $\Lambda(k)$ and $G(k)$ are linked through Equations (5) and (9). Therefore,

sizable corrections to $\rho_\Lambda$ can imply sizable corrections to $G(k)$, and, thus, to the Newton potential. This link is usually not considered when quantum corrections to the Newton potential are discussed [121].

(iii) Dynamical $\Delta t_{\mu\nu}$.

The modifications in $G(k)$ enter gravitational field Equation (13) twofold: first as a "static" modification of the prefactor of the usual matter energy density $\rho_M$ and second as a "dynamical" contribution through $\Delta t_{\mu\nu}$. In the WG-WSD limit in Equation (17), the latter contribution dominates over the former (more intuitive) contribution. To check the validity of the corresponding expansion (14), we can ignore the leading contribution to Equation (17) and recalculate the sensitivity (37) with the "static" $\Delta G/G_0$ term. This would result in a loss of sensitivity of 35 orders of magnitude. The same is true for conventional estimates like the one of Ref. [121], where no such 'dynamical' effect is considered.

(iv) Small skin depth.

Since the $\Delta t_{\mu\nu}$ contribution contains two spatial derivatives, the exponential form of $\rho_Q$ in Equation (35) leads to an additional enhancement near the boundary, and generally for $\delta_c \ll 1\,\mathrm{m}$. As mentioned, the experimental and theoretical uncertainty for this model may result in errors of several orders of magnitude. The skin effect is a particular feature of the experimental Casimir setup, for which it does not enter conventional estimates [121].

Note that the above four points are not independent ad hoc assumptions. They are, rather, natural consequences of the hypothesis (4).

### 3.2. Interpretation

An interpretation of the remarkable result (37) will be performed in inverted order of the items (iv)...(i) of the previous subsection.

(iv) It is well possible that our simulations of the conditions in the real experiment, in particular the skin depth relation (35), are an oversimplification. Independent of this fact, the boundary (37) strongly confines the parameters $\alpha C_1/(C_1 - C_3)$.

(iii) The result (37) would not hold if the $\Delta t_{\mu\nu}$ term was absent from modified field Equation (11). However, since this term is needed to restore diffeomorphism invariance, it can not be 'just absent'. It could, however, be replaced by a less minimal extension of GR, in which case Equation (37) would have to be recalculated for the particular non-minimal model.

(ii) It could happen that $\Lambda(k)$ is only very weakly linked to $G(k)$ in the infrared. In our parametrization, this possibility is contemplated for parameter points with $C_1 \ll C_1 - C_3$. From the SD perspective, this is would be an unusual scenario since, in typical benchmark scenarios, this is not the case (see Table A1).

(i) Finally, except for scenarios (iv)–(ii), there remains the possibility that Equation (37) provides a window into the unknown relation between the quantum world and cosmology/gravity. Due to the strength of the boundary (37), this will most likely, and under the assumption that WG-WSD represents a valid approach, allow experiments to gain insight on the CCP (2). This scenario will be discussed in the next subsection.

### 3.3. Back to the CCP

Could an experimental sensitivity of $\alpha \approx 10^{-30}$ teach us anything about the CCP (2)? The CCP arises from the ambition to predict $\rho_\Lambda$ in terms of $\rho_Q$ such that $\rho_\Lambda = \rho_\Lambda(\rho_Q)$. Without the loss of generality, we can write this ambition as

$$\rho_\Lambda = Y(\rho_Q)\rho_Q \,, \tag{41}$$

where we simply factored out a proportionality in $\rho_Q$. Phrased in this equation, the CCP is the statement that $Y_0$ is an extremely small number [see Equation (2)]. The fact that $Y_0$ is a small number, as measured in cosmology without additional Casimir energy contributions

(i.e., $\rho_Q = \rho_{Q,0}$), does not imply that it is actually a constant. It could be a function $Y = Y(\rho_Q)$ such that $Y_0 = Y(\rho_Q \to \rho_{Q,0})$. The quantum vacuum energy density $\rho_Q$, in turn, is for sure changing with additional small Casimir contributions

$$\rho_Q = \rho_{Q,0} + \beta \, \rho_C. \tag{42}$$

We have all reason to assume that the proportionality factor $\beta = 1$, but for the sake of being as general as possible, we will keep it arbitrary but $\leq 1$. A Casimir experiment gives the unique opportunity to actually realize tiny (linear) changes in $\rho_Q$ and, through the hypothesis (4), also changes in $\rho_\Lambda$. We define the dependence of the CCP on changes in the quantum energy density (through $\rho_C$) as

$$Y_0' \equiv \left. \frac{dY(\rho_Q)}{d\rho_C} \right|_{\rho_C = 0}. \tag{43}$$

Inserting Equations (2) and (4) into Equation (43) allows us to relate the observable $\alpha$ to $Y_0$ and $Y_0'$, namely

$$\alpha = Y_0' + \beta \, Y_0. \tag{44}$$

We already know that $Y_0$ [see (2)], which relates the cosmological constant $\Lambda_0$ to the vacuum energy density $\rho_{Q,0}$ in the absence of boundaries, is much smaller than the projected idealized sensitivity of $\alpha$ [see Equation (37)]. Thus, an upper bound on $\alpha$ would imply an upper bound on $Y_0'$, while the detection of a finite $\alpha$ would imply a non-vanishing value of $Y_0' = \alpha$, thereby directly yielding information about the dependence of $\Lambda$ on $\rho_C$, which was our working hypothesis for this study.

*3.4. Conclusions*

In this paper, we have explored the hypothesis (4) that cosmological energy density $\rho_\Lambda$ is subject to changes in quantum energy density in terms of the Casimir vacuum energy density $\rho_C$. The local nature of $\rho_C$ made it then inevitable to introduce scale dependence (SD) to the gravitational couplings such that both Newton's constant $G$ and the cosmological constant $\Lambda$ become dependent on the scale factor $k$ and could thus vary locally. SD, when minimally combined with diffeomorphism invariance, then led in the Weak Gravitational curvature and Weak SD (WG-WSD) limit to a modification of the gravitational potential (17) including non-trivial contributions from the vacuum (Casimir) energy.

This means that experiments, which are sensitive to both the gravitational force (21) and the Casimir force, have the potential to actually test our hypothesis. We derived a crude estimate Equation (37) for the achievable sensitivity in an upcoming experiment, and analyzed that such a test could give new insight into the cosmological constant problem and its possible origin in quantum fluctuations.

**Author Contributions:** Investigation—B.K., C.K., M.P. and R.I.P.S. All authors have read and agreed to the published version of the manuscript.

**Funding:** This work was supported by the Austrian Science Fund (FWF): P 34240-N.

**Data Availability Statement:** No new data were created or analyzed in this study. Data sharing is not applicable to this article.

**Acknowledgments:** We thank F. Intravaia for discussions on the intricacies of calculating $\rho_C$ and C. Laporte for valuable input concerning the beta functions in quantum gravity. We further thank H. Abele, C. Gooding, A. Padilla, I. A. Reyes, and H. Skarke for feedback and discussion.

**Conflicts of Interest:** The authors declare no conflict of interest.

## Appendix A. Parameters in Asymptotic Safety

Within a given model approach to quantum gravity, the coefficients $C_i$ cease to be free parameters. For the case of asymptotic safety coupled to matter, the beta functions for $g(k) = G(k) \cdot k^2$ and $\lambda(k) = \Lambda(k)/k^2$ are calculable [72,122–126] and given by

$$\beta_g = 2g + \frac{g^2}{6\pi} \cdot N_1, \tag{A1}$$

$$\beta_\lambda = -2\lambda + \frac{g}{4\pi}N_2 + \frac{g\lambda}{6\pi}N_3. \tag{A2}$$

Here, the coefficients $N_i$ are natural numbers, which depend on the matter content ($N_S$: number of scalars, $N_D$: number of Dirac particles, $N_V$: number of vector bosons) included in the asymptotic safety realization

$$N_1 = N_S + 2N_D - 4N_V - 46, \tag{A3}$$
$$N_2 = N_S - 4N_D + 2N_V + 2,$$
$$N_3 = N_S + 2N_D - 4N_V - 16.$$

By solving the beta functions and expanding the solutions in $k^2$, we find for the expansion parameters in Equations (9) and (10) that

$$C_1 = \frac{N_1}{12\pi}, \tag{A4}$$

$$C_3 = \frac{N_3}{12\pi}.$$

We will use these parameters for two benchmarks with matter. However, it is important to note that in the absence of matter (gravity only), most beta functions of the functional RG approach to gravity predict [127,128]

$$C_1 = C_3 \sim -\frac{15}{16\pi}. \tag{A5}$$

To put these parameters of the asymptotic safety approach in a phenomenological perspective, we discuss three benchmark scenarios in Table A1.

**Table A1.** Parameters of the three benchmark scenarios for renormalization group results; $B_1$ corresponds to pure gravity, $B_2$ is for the particle content of QED, and $B_3$ considers the particle content of the Standard Model.

|  | $B_1$ | $B_2$ | $B_3$ |
|---|---|---|---|
| $N_S$ | 0 | 0 | 4 |
| $N_D$ | 0 | 1 | 12 |
| $N_V$ | 0 | 1 | 12 |
| $C_1$ | $-15/(16\pi)$ | $-4/\pi$ | $-11/(2\pi)$ |
| $C_3$ | $-15/(16\pi)$ | $-3/(2\pi)$ | $-3/\pi$ |
| $C_1/(C_1 - C_3)$ | $\infty$ | 1.6 | 2.2 |

## Notes

[1] Earlier experiments on Casimir and van der Waals forces [24,25] (for a complete account see [26]) have suffered from severe uncertainties, for which their results can be considered to be rather qualitative instead of quantitative according to modern standards.

[2] In most configurations, the mode density is reduced and the Casimir force is attractive but theoretical [29–31] as well as experimentally tested counter examples exist [32–34] where the behavior is changed non-trivially either by material properties or geometry.

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
