# Peer review of "Vacuum Energy, the Casimir Effect, and Newton’s Non-Constant"

_universe, doi:10.3390/universe9110476_

Round 1

Reviewer 1 Report

Comments and Suggestions for Authors

In this paper, the authors discuss the potential impact of Casimir energy density on gravitational and cosmological "constants," which, within this framework, become scale dependent functions. Succinctly, the authors posit that both the Newton constant and the cosmological constant are not constants but rather depend on the local value of Casimir energy density. Subsequently, they employ generalized Einstein equations previously proposed in the literature for theories featuring scale-dependent variables $G$ and $\Lambda$  to compute corrections to macroscopic gravitational forces. The paper also explores possible experimental implications of their approach.

In my opinion the central assumption underpinning this paper appears somewhat ad hoc and lacks adequate motivation. Specifically, my concerns are as follows:

  1. The authors assert that the dependency of $G$ and $\Lambda$ on $ρ_C$ stems from the hypothesis in Eq.(4). This derivation seems rather evident, as they presume a linear relationship between $ρ_\Lambda$ and $ρ_C$, subsequently expanding $G(k)$ and $\Lambda (k)$ to linear order in $k^2$, leading to the conclusion that $k^2$ is proportional to $ρ_C$.
  2. There is no discussion regarding why one should regard $G$ and $\Lambda$ as functions of the Casimir energy density, as opposed to viewing the vacuum energy density as the source for the usual Einstein equations. The conventional approach within the context of quantum field theory in curved spacetimes would typically involve treating energy density as the source.
  3. Despite citing prior works on gravitational implications of Casimir energy, such as Ref.[43], the paper fails to address similarities and disparities with their own approach. Furthermore, there is no mention of the "Archimedes experiment" by E. Calloni et al., which aimed to validate the feasibility of measuring the interaction between vacuum fluctuations and gravity.
  4. The paper overlooks a discussion of the renormalization of vacuum energy. It appears that $ρ_C$ denotes the renormalized Casimir energy density, while, according to the description in Section 1, $ρ_{\Lambda_0}$ represents the non-renormalized vacuum energy. This raises questions about the compatibility of these concepts with Eq.(4).
  5. Section 2.2.2 presents arguments that are difficult to follow. The conventional approach to deriving scale-dependent effective actions usually involves integrating out degrees of freedom with momenta greater than $k$. The connection between the scale-dependent effective matter Lagrangian and the Casimir energy density in this section remains unclear.

In light of these concerns, I believe that the paper is not  suitable for publication.

Author Response

Responses to 1st referee's comments

We appreciate the comments and observations raised by the first referee. Below we give our detailed response:

Reviewer 2 Report

Comments and Suggestions for Authors

Referee's report on the paper "Vacuum energy, the Casimir effect,
and Newton's non-constant"
by B. Koch, C. Kaeding, M. Pitschmann, R.I.P. Sedmik

The authors discuss the cosmological constant problem and propose a possible
solution by establishing a relation of the cosmological constant with the
Casimir effect. The subject is of interest and the results obtained appear
to be new and correct, as far as I can see. However, I cannot recommend a
publication of the manuscript in its present form, a revision is needed.

1. To begin with, I am puzzled by the statement on page 1 (lines 17-19):
"Due to this prediction, it was believed..." "this long-held belief...". In
this relation, it is worthwhile to recall that physics is not a religion, but
an experimental science. The value of the deceleration parameter was measured
in numerous astrophysical observations, and the old measurements demonstrated
that it is positive which means a decelerating expansion. See, for example,
the books S. Weinberg, Gravitation and cosmology (Wiley, 1972) Ch. 14, Sec. 6,
p. 449, and also C. Misner, K.S. Thorn, J.A. Wheeler, Gravitation (Freeman,
1973) Sec. 29.4, p. 791.

2. In my opinion, the authors do not quite correctly write on page 2 (lines
66-67) that the Casimir effect was "experimentally confirmed nearly half a
century later [22]". Although it is certainly true that the modern measurements
started in 1997 after the work of Lamoreaux [22], such a statement produces a
misleading impression that before 1997 there was no experiments at all. However,
the Casimir effect was qualitatively and quantitatively (although, not quite
cleanly) confirmed much earlier, with the first measurements back to 1957-68.
For the discussion of the old experimental work, see the book of M. Bordag,
G.L. Klimchitskaya, U. Mohideen, V.M. Mostepanenko, Advances in the Casimir
Effect (Oxford University Press, 2009), Part III, Sec. 18, p. 513.

3. The whole presentation, includingg the motivation and the description and
discussion of the results obtained, is to a great extent incoherent, in my view.
Technically, the authors rely on the renormalization group (RG) approach in
the quantum field theory on curved manifolds. However, they tacitly assume
that there exists just one consistent RG approach, and they do not ever mention
there are different approaches in the literature. For example, the relevant
critical discussion of RG approaches can be found in the paper by I.L. Shapiro
et al [JCAP 01 (2005) 012] (see added note, and the reference list). Plainly
speaking, the manuscript is unreadable in its present form. The authors should
provide a coherent introduction, carefully explaining all the basic points of
their approach. In particular, the meaning of the scale k and scale-dependent
quantities should be clearly presented.

4. Minor issues are as follows.

-- The authors many times at various places speak about "Casimir effect from
electromagnetic fluctuations", however, in the main Sec. 2 the electromagnetism
is not mentioned at all. Moreover, as one knows, the Casimir effect can be also
derived for other fields (scalar, spinor, etc), so it is unclear to me why just
"electromagnetic fluctuations" are singled out in the text.

-- Please explain how the Casimir energy density \rho_C (which appears for the
first time on page 3, line 84) is related to the Casimir energy (3).

-- What is the "original cosmological energy density"? It is never explained
in a clear and unambiguous way.

-- I do not find any logical reason why the action (25) appears two pages after
the modified Einstein field equations (10) are presented without derivation.

Summarising, in view of the deficiencies above, a revision is required.

Author Response

Responses to 2nd referee's comments

We appreciate the comments and observations raised by the referee. Below we give our detailed response:

Reviewer 3 Report

Comments and Suggestions for Authors

The manuscript "Vacuum energy, the Casimir effect, and Newton’s non-constant", submitted for publication in Universe, addresses the fine-tuning problem of the cosmological constant by considering scale-dependent Newton's and cosmological constants.

The manuscript is interesting and well written, though there are a few points I would like to be addressed by the authors. They are listed below:

- Does the scale dependence of G involve only spatial coordinates or also temporal coordinates? If not, why is k independent of t? This should be better clarify in the text.

- What is the action which the field equations 10 come from? For the field equations to be consistent, they must come from a variational principle. For the benefit of the reader I suggest to expand the discussion in this direction. 

- I do not fully understand what is the main difference between the model discussed in this manuscript and a standard scalar-tensor model, where non-dynamical scalar fields are non-minimally coupled to the geometry and the scalar field itself plays the role of a scale-depending coupling constant. I suggest to introduce a brief discussion about scalar-tensor modifications of GR in the introduction.

- Eq. 13 is longer than the margin of the page.

- In Eq. 13, the authors write a static and spherically symmetric line element, assuming the Birkhoff theorem to hold for this model. However, the latter theorem is surely valid within the context of GR, but it must be proven to hold for different theories. The model considered in the manuscript consists of an extension of GR and it is not certain that the vacuum solution to the field equation is static. In my opinion, it would be worth to address this point in the revised version of the manuscript.

- This point is not closely related to the topic of the manuscript, but I wonder whether the field equations 10 admit analytic solutions in the background 13 (in vacuum). If so, the related discussion in the manuscript should be expanded in this direction.

- The field equations of the considered model are equivalent to those of f(R) gravity (see e.g. Eq. 10 of Ref. 0805.1726), with 1/f'(R) \equiv G(k) and [f(R) - R f'(R)] /(2 f'(R)) \equiv \Lambda(k). Is there a relation between this model and the f(R) gravity model? If so, it would be worth mentioning such a relation in the manuscript.

- Shouldn't the Energy-Momentum tensor in Eq. 14 simply read \rho_M/2 diag(-1,0,0,0), rather than \rho_M/2 diag(-1,0,0,0), for a pressureless fluid?

I believe that the authors should address the points outlined above before the manuscript can be accepted for publication in this journal.

Author Response

Responses to 3rd referee's comments

We appreciate the comments and observations raised by the referee. Below we give our detailed response:

Reviewer 4 Report

Comments and Suggestions for Authors

Excellent and interesting work, still hard to believe though that Casimir and gravitational effects will be disentangled experimentally...some minor formatting corrections for editors and maybe your arXiv version:

- formatting of "Eqns. (8), 9)" in lines 153, 163 and 172

- "of equation (10)" in line 181

- line break in eq. (13)

- extra bracket "Eqns. (8))" in line 233

Good luck with experiments!

Author Response

Responses to 4th referee's comments

We appreciate the comments and observations raised by the referee. Below we give our detailed response:

Round 2

Reviewer 2 Report

Comments and Suggestions for Authors

Referee's report on the revised paper "Vacuum energy, the Casimir effect,
and Newton's non-constant"
by B. Koch, C. Kaeding, M. Pitschmann, R.I.P. Sedmik

The authors had addressed all the points of my report and had amended the
manuscript in a satisfactory way. The paper is in a better shape now and can
be accepted for publication in Universe in its present form.

I have though some minor technical points which should be improved.

-- In my PDF copy of the revised manuscript, there are missing cross-references
for the papers cited, in particular, on line 53 (page 2) and lines 89, 123
(page 3). Please improve cross-referencing.

-- The name of the first author in the reference [89] is misspelled. The correct
name sounds Damour.

Author Response

We thank the referee,  the

reference-linking and the author name were corrected.

Reviewer 3 Report

Comments and Suggestions for Authors

The authors have addressed the concerns pointed out in my previous review. Therefore I recommend the manuscript for publication in this journal.

Author Response

We thank the reviewer and the editor.